# Effect of the N-hexanoyl-L-homoserine Lactone on the Carbon Fixation Capacity of the Algae–Bacteria System

**DOI:** 10.3390/ijerph20065047

**Published:** 2023-03-13

**Authors:** Lei Liao, Bin Chen, Kaikai Deng, Qiang He, Guijiao Lin, Jinsong Guo, Peng Yan

**Affiliations:** 1College of Environment and Ecology, Chongqing University, Chongqing 400045, China; 2Key Laboratory of Ecological Environment of Three Gorges Reservoir Area, Ministry of Education, Chongqing 400045, China

**Keywords:** algae–bacteria system, C_6_-HSL, inorganic carbon, CCM model, carbon fixation capacity

## Abstract

Algae–bacteria systems are used widely in wastewater treatment. N-hexanoyl-L-homoserine lactone (AHL) plays an important role in algal-bacteria communication. However, little study has been conducted on the ability of AHLs to regulate algal metabolism and the carbon fixation ability, especially in algae–bacteria system. In this study, we used the *Microcystis aeruginosa + Staphylococcus ureilyticus* strain as a algae–bacteria system. The results showed that 10 ng/L C_6_-HSL effectively increased the chlorophyll-a (Chl-a) concentration and carbon fixation enzyme activities in the algae–bacteria group and algae group, in which Chl-a, carbonic anhydrase activity, and Rubisco enzyme increased by 40% and 21%, 56.4% and 137.65%, and 66.6% and 10.2%, respectively, in the algae–bacteria group and algae group, respectively. The carbon dioxide concentration mechanism (CCM) model showed that C_6_-HSL increased the carbon fixation rate of the algae–bacteria group by increasing the CO_2_ transport rate in the water and the intracellular CO_2_ concentration. Furthermore, the addition of C_6_-HSL promoted the synthesis and secretion of the organic matter of algae, which provided biogenic substances for bacteria in the system. This influenced the metabolic pathways and products of bacteria and finally fed back to the algae. This study provided a strategy to enhance the carbon fixation rate of algae–bacteria consortium based on quorum sensing.

## 1. Introduction

The greenhouse effect, caused by increasing concentrations of CO_2_, CH_4_, and N_2_O in the atmosphere, is a major problem around the world [1]. Carbon dioxide (CO_2_) contributes to about 76% of all greenhouse gases [2]. Among the existing strategies developed to fix CO_2_, the use of algae to reduce atmospheric CO_2_ is a widely accepted fixation strategy [3]. As a primary producer, algae can fix atmospheric CO_2_ in cells through photosynthesis and generate biodiesel, protein, and other substances with economic value through a series of transformations [4,5,6].

Algae–bacteria systems receive more attention because of the mutually beneficial relationship of compounds metabolized by algae and bacteria in carbon fixation strategies [7]. Bacteria utilize the organic compounds released by algal photosynthesis, while algae assimilate the CO_2_ and HCO_3_^−^ produced by bacteria, resulting in a symbiotic relationship between algae and bacteria [8]. With the recycling of the compounds, the algae–bacteria system can treat wastewater with low energy consumption, high efficiency, and high economic benefit [9,10,11]. Furthermore, bacteria can affect the carbon fixation process of algae with its secretions. Like many organisms, algae depend on bacteria for ultimate exogenous sources of cobalamin (vitamin B_12_), thiamine (vitamin B_1_), and biotin (vitamin B_7_) as co-factors for B_12_-dependent methionine synthase to enhance their growth [12]. Some special bacteria can produce indole-3-acetic acid (IAA) to promote algal growth [13,14,15], stimulate lipid production, and optimize fatty acid composition [16]. Meanwhile, the change in pH and nutrient concentration caused by bacteria can also affect algal metabolism [17,18].

The exchange of information between cells in an algae–bacteria system is conducted by quorum sensing, in which acyl-homoserine lactones (AHLs) are important signal molecules in the intercellular communication of gram-negative bacteria [19]. AHLs can affect bacteria activities, such as biofilm formation [20,21,22], pigment synthesis [23], and antibiotic synthesis [24]. Therefore, they play an important role in wastewater treatment technology [25,26]. AHLs can directly regulate not only the physiological activities of bacteria [27] but also the metabolism of algae (e.g., lipid synthesis) [28]. Some studies showed that 400 nmol/L N-hexanoyl-L-homoserine lactone (C_6_-HSL) was effective in promoting the accumulation of lipid concentration in algae cells in the mixed culture system of *Chlorella vulgaris* and *Chlamydomonas* [29]. The effect of C_6_-HSL on algae cells was mainly reflected in DNA replication and lipid synthesis, and it was found that C_6_-HSL could up-regulate the key enzyme activity of lipid synthesis and down-regulate the key enzyme activity of DNA in *Chlorella vulgaris* [28].

Inorganic carbon concentration is an important factor in limiting the algal carbon fixation rate. Thus, algae activate the unique CO_2_ concentration mechanism (CCM) to promote the dehydration of HCO_3_^−^ inside and outside the cell under the catalyzation of carbonic anhydrase (CA) to satisfy the algae’s CO_2_ requirements [30,31]. Interestingly, the CCM can only be induced in low-CO_2_ (LC, ~0.04% CO_2_) or very low–CO_2_ (VLC, <0.02% CO_2_) conditions. Maybe the high CO_2_ (HC, >1% CO_2_) significantly suppresses the activity of CAs [32]. The concentrations of CO_2_ and HCO_3_^−^ in the water can regulate the CCM mechanism and thus affect the physiological activities of algae [33]. Thus, the concentration of inorganic carbon in the water will affect the CCM mechanism and the carbon fixation capacity of algae [28].

Inorganic carbon (Ci) in the form of CO_2_ has an acidic pH value, while the overwhelming majority of Ci is in the form of HCO_3_^−^ at an alkaline pH, which makes the algae suffer from dramatic changes in the supply of dissolved inorganic carbon (DIC) [32]. These changes are more obvious in the algae–bacteria system because of the changes in pH caused by bacteria [34]. However, little study has been conducted on the ability of the algae–bacteria system to change the supply of DIC, which affects the algal carbon fixation process, especially with the addition of AHLs.

This study investigates the effect of the representative C_6_-HSL in AHLs on the carbon fixation capacity of an algae–bacteria system that is constructed with the bloom-dominant algae *Microcystis aeruginosa* and the bacteria *Staphylococcus ureilyticus* isolated from the soil of water-level fluctuation zone (WLFZ) in Gaoyang Lake of Pengxi River in Three Gorges Reservoir. Using a modified Fridlyand model [35], the mechanism of the effect of AHLs on algal carbon fixation rate is revealed.

## 2. Materials and Methods

### 2.1. Experimental Design

The experiment was divided into two parts. The first part of the experiment (Exp. I) was conducted to investigate the effect of the signal molecule C_6_-HSL in different concentrations on the growth of the algae–bacteria group. The specific steps are as follows: The expanded cultured *Microcystis aeruginosa* and *Staphylococcus ureilyticus* strains were collected by centrifuge (5000 rpm, 30 min) and resuspended in a 1 L flask with sterilized BG11 medium [36] (121 °C, 30 min). Both the initial cell densities of algae and bacteria were set at 1 × 10^5^ cells/L. C_6_-HSL was added in the middle of the logarithmic growth phase of algae (day 10 of the cultivation cycle), the concentration gradient was set as 0, 5, 10, 50, 500, and 1000 ng/L, and duplicate samples were set in each group. Therefore, the experiment was divided into two phases: In Phase I, C_6_-HSL was not added in the early stage of the experiment (days 2–10); in Phase II, C_6_-HSL was added in the middle and late stages of the experiment (days 10–14). The indicators measured in Exp. I were chlorophyll-a (Chl-a) concentration and the activities of the CA enzyme and Rubisco enzyme in the carbon fixation process.

The second part of the experiment (Exp. II) investigated the effect of C_6_-HSL on the change of carbon concentration in algae cells in the algae–bacteria system based on the CCM model. The C_6_-HSL concentration was determined in Exp. I, which was the optimal growth in the algae–bacteria group. The procedures in Exp. II were the same as in Exp. I. The indicators measured in Exp. II were Chl-a, maximum photochemical quantum yield (Fv/Fm), TIC, TOC, and the activities of the CA enzyme and Rubisco enzyme in the carbon fixation process.

### 2.2. Experimental Materials

*Microcystis aeruginosa* (FACHB-905) was provided by Freshwater Algae Culture Collection at the Institute of Hydrobiology, Chinese Academy of Science. The *Staphylococcus ureilyticus* strain CK27 was isolated from the soil of the water-level fluctuation zone (WLFZ) in Gaoyang Lake of Pengxi River [37] in Three Gorges Reservoir and identified after high throughput sequencing. C_6_-HSL was purchased from Aladdin Reagent Company (Shanghai, China).

### 2.3. Experimental Methods

#### 2.3.1. Inoculation of Algae–Bacteria System

*Staphylococcus ureilyticus* strain CK27 was cultured in a vibrator (ZD-85A; Langyue, China) at 60 rpm and 30 °C for 24 h. Then, 10 mL each of fresh bacteria and expended *Microcystis aeruginosa* were taken for centrifugation (10,000 rpm for 5 min) to concentrate them for inoculating in flasks with 800 mL BG11 culture medium. The algae:bacteria cell density was 10^5^:10^5^. The illumination condition was 40 μmol/m^2^ with 12 h:12 h light to dark cycles. The temperature was set at 25 ± 1 °C.

#### 2.3.2. Detection of Growth Characteristics of Algae–Bacteria Systems

The Fv/Fm of algae was determined by taking the algal and algae–bacteria mixture samples, placing them in a dark environment for 30 min, and using Aquapen (Ap100, Drasov, Czech Republic) to detect.

The acetone extraction method was used to detect the Chl-a concentration of algae. First, 10 mL each of the algae and algae–bacteria mixture samples were centrifuged at 10,000 rpm for 10 min. After that, the supernatant was poured off, and 5 mL of 90% acetone solution was added and then placed in a refrigerator at 4 °C overnight. Last, the absorbance of samples was measured at 630, 645, 663, and 750 nm. The Chl-a of samples was calculated by using Equation (1).
(1)$$\frac{\left(11.64\times \left({\mathrm{OD}}_{663}-{\mathrm{OD}}_{750}\right)-2.16\times \left({\mathrm{OD}}_{645}-{\mathrm{OD}}_{750}\right)+0.1\times \left({\mathrm{OD}}_{630}-{\mathrm{OD}}_{750}\right)\right)\times V_{1}}{V_{2}}\;$$
where V_1_ is the volume of algal fluid and V_2_ is the volume of 90% acetone solution added.

#### 2.3.3. Carbon Concentration and Morphology Testing

After taking 10 mL of the algal and 10 mL of the algae–bacteria mixture samples from each group and centrifuging at 10,000 rpm for 10 min, the supernatants were each transported to a new centrifuge tube. After the supernatants were filtered by a 0.45 μm filter membrane, the filtered supernatants were placed in glass bottles. The total organic carbon (TOC) and total inorganic carbon (TIC) of each group were measured by a TOC instrument (Elementarvario TOC cube, Langenselbold, Germany).

The concentration of different forms of carbon (C) in TIC was subsequently calculated using Equations (2)–(4) [38].
(2)$$\;C({\mathrm{CO}}_{2})=\mathrm{TIC}-C\left({\mathrm{HCO}}_{3}^{-}\right)-\;C\left({\mathrm{CO}}_{3}^{2-}\right)\;$$
(3)$${\mathrm{HCO}}_{3}^{-}=\frac{\mathrm{TIC}\times K_{1}\times H^{+}}{{\left(H^{+}\right)}^{2}+K_{1}\times H^{+}+K_{1}\times K_{2}}\;$$
(4)$${\;\mathrm{CO}}_{3}^{2-}=\frac{K_{2}\times {\mathrm{HCO}}_{3}^{-}}{H^{+}}$$
where $C\left({\mathrm{CO}}_{2}\right),\;C\left({\mathrm{HCO}}_{3}^{-}\right),\;C\left({\mathrm{CO}}_{3}^{2-}\right)$ is the concentration of ${\mathrm{CO}}_{2}$, ${\mathrm{HCO}}_{3}^{-}$ and ${\mathrm{CO}}_{3}^{2-}$ in the water (mmol/L); TIC is the concentration of TIC in the water (mmol/L);$\;H^{+}$ is the concentration of $H^{+}$ in the water (mmol/L); and $K_{1},{\;K}_{2}$ are equilibrium constants of the first and second carbonate ionization, respectively.

#### 2.3.4. Carbonic Anhydrase and Rubisco Enzyme Activities

At the end of the experiment (day 14), samples were taken to determine algal carbonic anhydrase and Rubisco enzyme activities after cell fragmentation:

CA enzyme was measured using the electrode method, and unit enzyme activity was calculated using Equation (5) [39].
(5)$$\mathrm{EU}=10\times \left(\frac{T_{0}}{T}-1\right)$$
where $T_{0}$ is the time to reduce the pH from 8.3 to 6.3 for the blank sample and T is the time to reduce the pH from 8.3 to 6.3 for the algal solution or algae–bacteria mixture solution.

Rubisco enzyme activity was measured using a ribulose diphosphate carboxylase (Rubisco) kit (Suzhou GRS Biotechnology Co., Ltd., Suzhou, China).

#### 2.3.5. CCM Model

The carbon cycle within the algae cell was calculated using Fridlyand’s improved CCM calculation model based on the Reinhold model [35]. The model is shown in Figure 1. The model divides the flow of inorganic carbon into the following processes: (1) Through the plasmalemma: reversible transport of HCO_3_^−^ from the cytoplasm to the medium, diffusion of CO_2_ from the medium into the cytoplasm. (2) In the cytoplasm: conversion of CO_2_ into HCO_3_^−^ by the CA-like entity, spontaneous interconversion between CO_2_ and HCO_3_^−^. (3) In the carboxysomes: diffusion of HCO_3_^−^ from the cytoplasm into the carboxysomes, diffusion of CO_2_ from the carboxysomes into the cytoplasm, CA catalyzed interconversion between CO_2_ and HCO_3_^−^, fixation of CO_2_. The detailed parameters of CCM model used in this paper was shown in Table A1.

#### 2.3.6. Statistical Analysis

Data were plotted using Origin2018. One-way significance analysis was conducted in IBM SPSS Statistics 26 (ANOVA).

## 3. Results

### 3.1. Effects of C_6_-HSL on the Growth of Algae–Bacteria System in Exp. I

The effect of C_6_-HSL with different concentrations on Chl-a in the algae–bacteria system is shown in Figure 2. Before the addition of C_6_-HSL (days 2–10), the Chl-a of the algae–bacteria group started to increase significantly on day 4, while the Chl-a of the algae group started to increase only on day 8. The concentration of Chl-a of the algae–bacteria group was higher than that of the algae group during the growth process. After the addition of C_6_-HSL with different concentrations, only the 10 ng/L C_6_-HSL treatment could promote Chl-a concentration in the algae group, and the Chl-a concentration increased by 21% compared with the 0 ng/L C_6_-HSL treatment in the algae group on day 14. Compared with the 0 ng/L treatment in the algae–bacteria group, C_6_-HSL promoted the Chl-a concentration of each algae–bacteria group, where the Chl-a concentration of the 10 ng/L C_6_-HSL treatment was the highest (a 40% increase compared to the 0 ng/L C_6_-HSL group).

The effect of different concentrations of C_6_-HSL on the activities of important enzymes in the carbon fixation process is shown in Figure 3. The Rubisco enzyme activity gradually increased with increasing C_6_-HSL in all groups (except for the 500 ng/L group) in the algae group treated with C_6_-HSL (algal+C_6_-HSL group). It is noteworthy that only the 10 ng/L C_6_-HSL group was effective in enhancing the CA enzyme activity of the algae in the algae–bacteria group treated with C_6_-HSL (algae–bacteria+C_6_-HSL). Under the treatment of 10 ng/L C_6_-HSL, the CA enzyme and Rubisco enzyme activities of the algae–bacteria+C_6_-HSL group were increased by 56.4% and 66.6%, respectively, compared with the algae–bacteria group without C_6_-HSL. Therefore, the appropriate concentration of C_6_-HSL was 10 ng/L, which could increase the Chl-a concentration and the activities of the CA enzyme and Rubisco enzyme.

### 3.2. Effects of C_6_-HSL on the Growth of Algae–Bacteria System in Exp. II

According to Exp. I (Section 3.1), we chose 10 ng/L as the optimal C_6_-HSL concentration and used it in Exp. II. The changes in Chl-a concentration of algae under the treatment of 10 ng/L C_6_-HSL are shown in Figure 4a. In Phase I, the Chl-a concentration of algae–bacteria group was higher than that of the algae group. After adding C_6_-HSL (Phase II), the Chl-a concentrations of the algae–bacteria+C_6_-HSL group and the algal+C_6_-HSL group were increased by 5% and 23%, respectively. At the end of the experiment, the Chl-a concentration of the algae–bacteria+C_6_-HSL group reached 0.83 mg/L, which was higher than that of the algal+C_6_-HSL group at 0.73 mg/L. This was in agreement with the results mentioned in Section 3.1.

The changes in Fv/Fm in the algal and algae–bacteria groups with the treatment of 10 ng/L C_6_-HSL are shown in Figure 4b. It can be seen that without adding C_6_-HSL, the Fv/Fm in the algae–bacteria group was always lower than the algae group from day 4 to 14. With the addition of C_6_-HSL, it was found that C_6_-HSL could effectively promote the Fv/Fm of algae in the algae–bacteria+C_6_-HSL group, and it was increased by 5.8% on day 14.

The effects of 10 ng/L C_6_-HSL on the activities of the CA enzyme and Rubisco enzyme are shown in Figure 5a,b. The CA enzyme and Rubisco enzyme activities of the algae–bacteria group were higher than those of the algae group, and the Rubisco enzyme increased by 233%, and the CA enzyme increased by 23.1% after adding C_6_-HSL. This was in agreement with the results mentioned in Section 3.1.

### 3.3. Changes in TIC and TOC Concentrations in the Algae–Bacteria System

The changes in TIC concentrations in the algae–bacteria group with the treatment of 10 ng/L C_6_-HSL are shown in Figure 6a. In Phase I, the TIC of the algae group decreased slowly from day 2 to day 6 and increased after day 10. TIC concentrations in the algae–bacteria group fluctuated between 5.12 and 5.68 mg/L during the cultivation cycle. With the addition of C_6_-HSL (Phase II), the TIC concentrations of the algae+bacteria+C_6_-HSL and algal+C_6_-HSL groups were increased by 48.9% and 45.5% on day 14, respectively.

The variation of TOC concentrations in the algae–bacteria system is shown in Figure 6b. In Phase I, the TOC concentrations of the algae–bacteria group and the algae group fluctuated in the first 10 days, and the TOC concentration of the algae–bacteria group was lower than that of the algae group. With the addition of C_6_-HSL, the TOC concentrations of the algae+bacteria+C_6_-HSL and algae+C_6_-HSL groups increased by 85.3% and 26.7%, respectively.

The effects of C_6_-HSL on the concentration of different forms of TIC in the algal and algae–bacteria groups are shown in Figure 6c,d. The CO_2_ concentrations of TIC in both the algal and algae–bacteria groups decreased gradually at the beginning of the experiment and reached the lowest values on day 10. The TIC was mainly composed of HCO_3_^−^ and CO_3_^2−^, then recovered to the previous level on day 12. With the addition of C_6_-HSL, the CO_2_ concentrations in the TIC of the algal+C_6_-HSL and algae+bacteria+C_6_-HSL groups increased by 94% and 634%, respectively.

### 3.4. The carbon Fixation Ability of Algae–Bacteria System Based on the CCM Model

The effects of 10 ng/L C_6_-HSL on the CO_2_ transport rate in the water of the algae and algae–bacteria groups calculated by the CCM model are shown in Figure 7. From day 2 to 10, the CO_2_ transport rate in the water of the algae group was higher than that of the algae–bacteria group. It is worth noting that the CO_2_ transport rate in the water of the algae–bacteria group showed a negative value on day 10. The CO_2_ transport rate in the water of the algae–bacteria group was increased with the addition of C_6_-HSL (increased by 31.0%), while that of the algae group decreased.

The intracellular CO_2_ concentrations of algae in the algae and algae–bacteria groups calculated by the CCM model are shown in Figure 8. The intracellular CO_2_ of the algae–bacteria and algae groups gradually reduced from day 2 to 10 and then recovered to the previous level on day 12. It is noteworthy that the algae–bacteria group had a lower intracellular CO_2_ concentration compared to the algae group from day 2 to 10, while it was higher than the algae group on day 12. With the addition of C_6_-HSL, the algae+bacteria+C_6_-HSL group showed an increase in intracellular CO_2_ concentration.

The carbon fixation rates of algae in the algal and algae–bacteria groups calculated by the CCM model are shown in Figure 9. It can be seen that the carbon fixation rate of algae–bacteria group was lower than that of the algae group from day 2 to 10, and all groups reached the lowest value on day 10 and recovered to the previous level on day 12. It is worth noting that the carbon fixation rate of algae–bacteria was higher than that of the algae group after day 12. The carbon fixation rate of the algae+bacteria+C_6_-HSL group and algal+C_6_-HSL group increased by 2% and 1%, respectively, after the addition of C_6_-HSL.

## 4. Discussion

It has been reported that AHLs, quorum-sensing signal molecules, are used to enhance the wastewater treatment performance of an algae–bacteria system [40,41]. C_6_-HSL is a representative compound of AHLs, which influences the biofilm formation and physiological activities of bacteria by mediating the quorum sensing of gram-negative bacteria [42,43]. *Microcystis aeruginosa* used in this experiment is cyanobacteria (belonging to prokaryote), which is the same as gram-negative bacteria. Therefore, it may regulate its metabolism with AHLs. In this paper, we found that C_6_-HSL could effectively increase the Chl-a concentration and improve the key carbon fixation enzyme activity of algae. It proved that C_6_-HSL could directly affect algae growth, just like gram-negative bacteria. The most obvious effect of C_6_-HSL on algae growth was observed when the concentration was 10 ng/L, which indicated that the algae growth could be improved under the low concentration of C_6_-HSL. The Fv/Fm reflected the potential maximum photosynthetic capacity of the algae cells. The increase in Fv/Fm values caused by the addition of C_6_-HSL in the algae group explained why C_6_-HSL could promote the accumulation of Chl-a in the algae cells.

Moreover, the changes in TIC and TOC in the system also proved that C_6_-HSL could act directly on algae. In this study, we found that in the algae+C_6_-HSL group, TIC and TOC concentrations increased with the addition of C_6_-HSL. For the algae system, the main sources of TIC were the culture medium and the diffusion of CO_2_. The consumption of TIC was caused by the photosynthesis of the algae with the secretion of TOC into water. Thus, the increase of TOC indicated that C_6_-HSL could promote the algal carbon fixation process to synthesize and secrete more organic substances. The increased carbon fixation rate calculated by the CCM model also proved this. Meanwhile, more production of TOC meant more consumption of TIC. Therefore, more CO_2_ diffused from the atmosphere into the water to satisfy the increasing TIC demand. The intracellular CO_2_ concentration of algae increasing with the addition of C_6_-HSL also indicated algae largely absorbed CO_2_ from water into the cell, owing to the increasing DIC demand. When the CO_2_ diffusion rate was greater than the TIC consumption rate, the TIC concentration increased in the algae+C_6_-HSL group.

The differences between the algae and algae–bacteria groups without the addition of C_6_-HSL were mainly caused by bacteria. In this study, we found that the Fv/Fm value of the algae–bacteria group was lower than that of the algae group during the growth. This suggested that the addition of bacteria may cause environmental stresses (e.g., nutrients, pH) on the algae, which reduces the Fv/Fm value of algae [42,44]. Meanwhile, the CO_2_ concentration tended to be 0 on day 10 only in the algae–bacteria group. This might be caused by the compounds secreted by bacteria changing the pH value in the system. The changes in pH value made the CO_2_ concentration tend to be 0 in the algae–bacteria group, which limited the growth activity of algae [45,46]. Although the algae can activate the CCM mechanism to use HCO_3_^−^ as a carbon source, the conversion of HCO_3_^−^ into CO_2_ catalyzed by the CA enzyme requires energy consumption. Therefore, the algae redistribute the energy obtained from photosynthesis, which reduces the energy used in the carbon fixation process and leads to the decline of the carbon fixation rate [47]. In this study, we found that algae tended to metabolize the stored substances in the cells to obtain the energy for growth and then released CO_2_ into the water to change the pH value in the system, restoring the CO_2_ concentration to the previous level and rationalizing the energy distribution. The negative CO_2_ transport rate in the water of the algae–bacteria group on day 10 proved this. After the algae’s self-regulation, the Chl-a concentration of the algae–bacteria group was higher than the algae group from day 12 to 14. This indicated that the bacteria could promote the Chl-a synthesis of algae after algae adaption.

The CCM model also explains why bacteria can promote algal growth. The changes in CO_2_ concentration caused by bacteria also affect the intracellular CO_2_ concentration of algae. In this study, we found that the intracellular CO_2_ concentration of the algae and algae–bacteria groups gradually decreased from day 2 to day 10 and reached the lowest value on day 10. The decrease in intracellular CO_2_ concentration can be explained in two ways: Firstly, the decrease in CO_2_ concentration in the system led to the decrease of CO_2_ transport rate in the water, and thus the CO_2_ concentration of the cytoplasm decreased gradually; secondly, CO_2_ was fixed continuously from the cytoplasm to the carboxysomes. Because the CO_2_ concentrations in the carboxysomes were positively correlated with the carbon fixation rates of algae, the carbon fixation rates of algae in the algae and algae–bacteria groups were lowest on day 10 [48]. Subsequently, the algae and algae–bacteria groups returned to normal levels of CO_2_ in the carboxysomes and cytoplasm (day 12–14) because of the self-regulating by algae.

The mechanism of the effect of C_6_-HSL on the algae–bacteria system is shown in Figure 10. The bacteria used in this experiment were gram-positive bacteria, and the literature reports that C_6_-HSL cannot mediate their quorum-sensing [49,50]. However, C_6_-HSL can increase the Fv/Fm value of algae to improve the Chl-a synthesis process and promote carbon fixation to produce and secrete organic compounds into water, as mentioned before. In this paper, we found that the TOC concentration in the algae+bacteria+C_6_-HSL group was increased and higher than that of the algal+C_6_-HSL group with the addition of C_6_-HSL. It can be explained in two ways: Firstly, C_6_-HSL can directly promote the carbon fixation process of algae; secondly, more organic compounds produced by algae with the addition of C_6_-HSL can improve the bacteria metabolism, which can convert them to inorganic compounds to enhance the carbon fixation process of algae. Furthermore, inorganic carbon existed as CO_2_ because of the decreased pH value caused by algae. The abundant CO_2_ concentration increases the carbon fixation rate of algae. In this study, we found that the algae+bacteria+C_6_-HSL group showed an increase in intracellular CO_2_ concentration of algae with the addition of C_6_-HSL. It was caused by the increase in carbon fixation rate. The increase in algae carbon fixation rate led to an increase in intracellular CO_2_ consumption rate, causing algae to absorb CO_2_ from the outside to satisfy their need; the increase in the carbon fixation rate of algae promoted the cells to transfer CO_2_ from the cytoplasm to the carboxysomes, and the CA enzyme catalyzed the conversion of HCO_3_^−^ to CO_2_ in the cells, thus increasing the CO_2_ concentration in the carboxysomes to meet the needs of algae. After the addition of C_6_-HSL, the activity of the CA enzyme and Rubisco enzyme of the algae–bacteria+C_6_-HSL group increased on day 14. This proved the increase in carbon fixation rate and intracellular CO_2_ concentration. Thus, the mechanism of the effect of C_6_-HSL on the algae in the algae–bacteria system was revealed: Firstly, C_6_-HSL could directly enhance algal carbon fixation ability to secrete more organic compounds into the water; secondly, the abundant organic compounds were utilized by bacteria to produce inorganic compounds; thirdly, bacteria could change the pH value to increase CO_2_ concentration; finally, the increased CO_2_ concentration and abundant inorganic compounds enhanced the algal carbon fixation ability.

## 5. Conclusions

In this study, we proved that C_6_-HSL promoted the growth of Microcystis aeruginosa, which is a kind of prokaryote similar to gram-negative bacteria. The mechanism of the effect of C_6_-HSL on the algae in the algae–bacteria system was C_6_-HSL mainly providing biogenic substances to the bacteria in the system by affecting the synthesis and secretion of the algae’s TOC and influencing the metabolic pathways and products of the bacteria, which eventually fed back to the algae to promote their metabolism. This provides a new way to regulate the physiological activity of the algae–bacteria system by using C_6_-HSL to enhance the carbon fixation of the algae. However, the mechanism of the effect of C_6_-HSL on *Microcystis aeruginosa* in the algae–bacteria system needs to be investigated further in future study.

## Figures and Tables

**Figure 1 ijerph-20-05047-f001:**
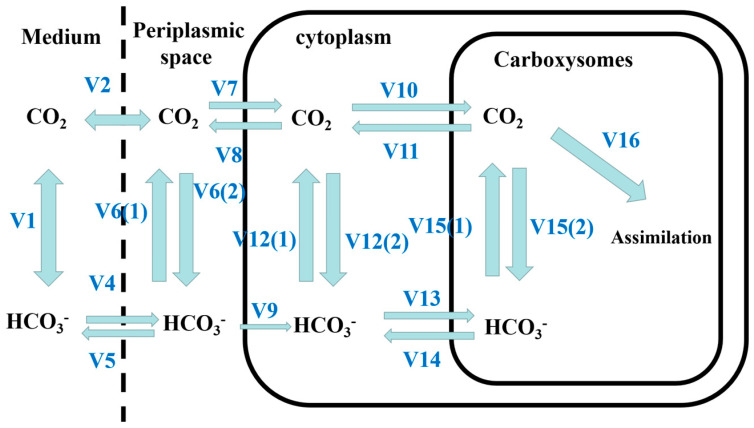
Fridlyand calculation model.

**Figure 2 ijerph-20-05047-f002:**
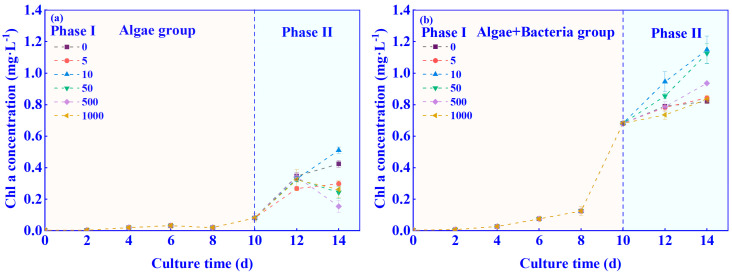
Effect of different concentrations C_6_-HSL on Chl-a concentration in (**a**) algae group and (**b**) algae–bacteria group. Numbers mean the concentration of C_6_-HSL (ng/L). Phase I is without the addition of C_6_-HSL, and Phase II is with the addition of C_6_-HSL.

**Figure 3 ijerph-20-05047-f003:**
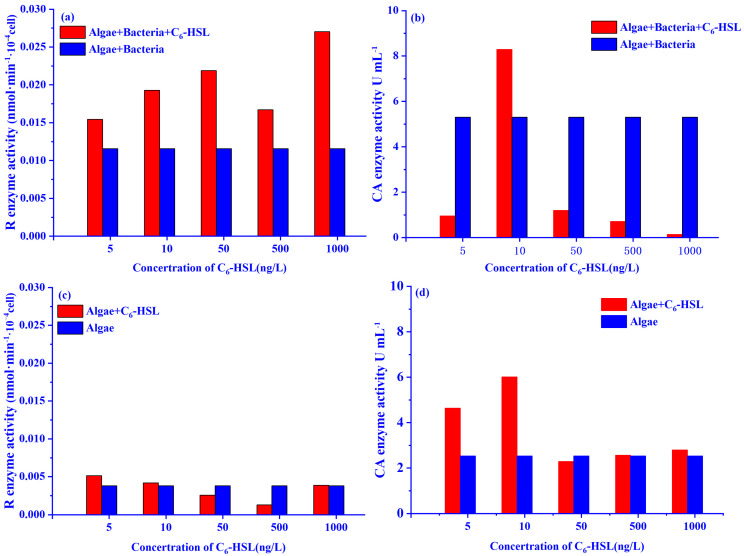
Effect of different concentrations C_6_-HSL on the activity of the (**a**,**c**) Rubisco (R) enzyme and (**b**,**d**) CA enzyme of algae in algal and algae–bacteria groups, respectively.

**Figure 4 ijerph-20-05047-f004:**
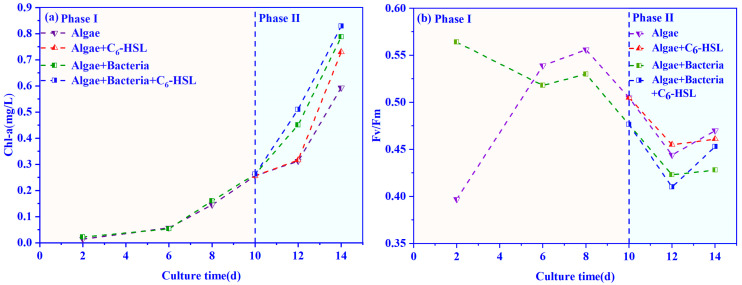
Effect of 10ng/L C_6_-HSL on (**a**) Chl-a and (**b**) Fv/Fm of algae group and algae–bacteria group. Algal+C_6_-HSL and algae+bacteria+C_6_-HSL are algae group and algae–bacteria group treated with C_6_-HSL, respectively. Phase I is without the addition of C_6_-HSL, and Phase II is with the addition of C_6_-HSL.

**Figure 5 ijerph-20-05047-f005:**
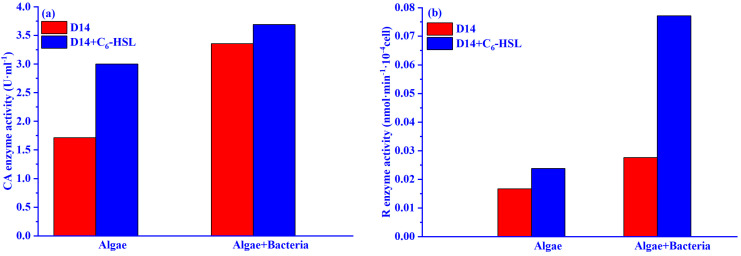
Effect of 10ng/L C_6_-HSL on (**a**) CA and (**b**) Rubisco of algae group and algae–bacteria group. Algal+C_6_-HSL and algae+bacteria+C_6_-HSL are algae group and algae–bacteria group treated with C_6_-HSL, respectively. Phase I is without the addition of C_6_-HSL, and Phase II is with the addition of C_6_-HSL.

**Figure 6 ijerph-20-05047-f006:**
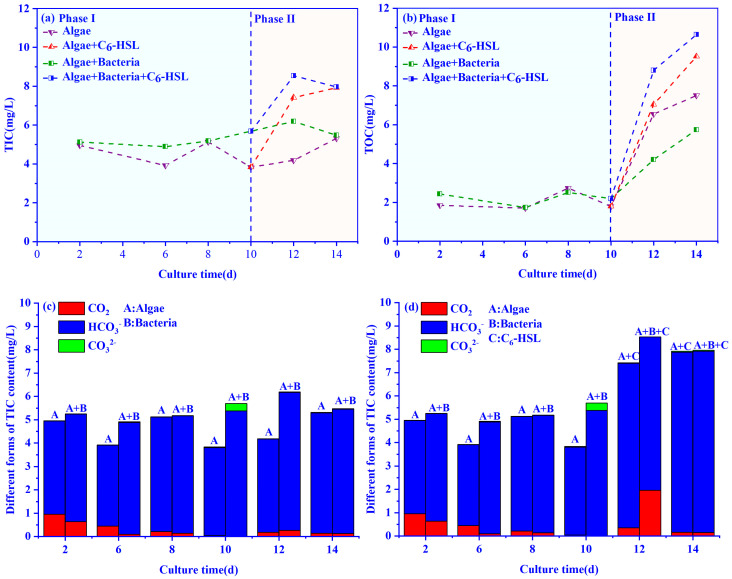
Effect of 10ng/L C_6_-HSL on (**a**) TIC and (**b**) TOC of algae group and algae–bacteria group. Algae+C_6_-HSL and algae+bacteria+C_6_-HSL are algae group and algae–bacteria group treated with C_6_-HSL, respectively. Phase I is without the addition of C_6_-HSL, and Phase II is with the addition of C_6_-HSL. Different forms of TIC concentrations of algae group and algae–bacteria group (**c**) without or (**d**) with the addition of C_6_-HSL. A is algae group, A + B is algae–bacteria group, and A + C and A + B + C are algae group and algae–bacteria group treated with C_6_-HSL, respectively.

**Figure 7 ijerph-20-05047-f007:**
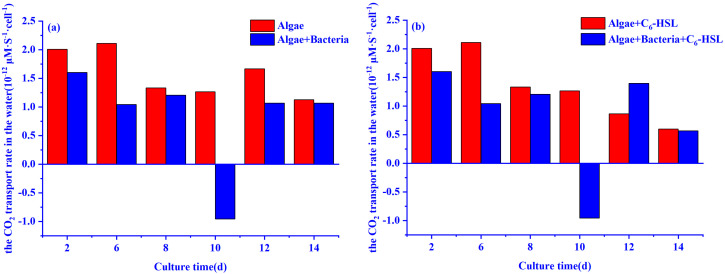
The CO_2_ transport rate in the water of algae and algae–bacteria groups (**a**) without or (**b**) with the addition of C_6_-HSL. Algae+C_6_-HSL and algae–bacteria+C_6_-HSL are algae group and algae–bacteria group treated with C_6_-HSL, respectively.

**Figure 8 ijerph-20-05047-f008:**
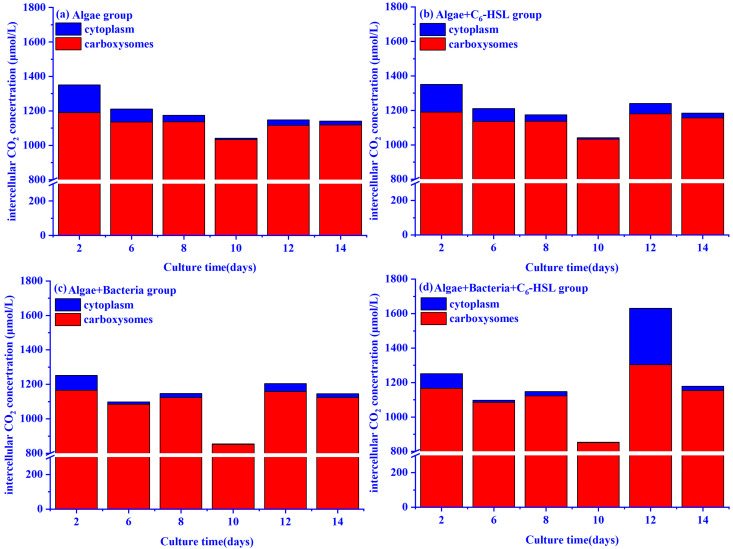
Effect of 10 ng/L C_6_-HSL on the intracellular (cytoplasm and carboxysomes) CO_2_ concentrations of algal and algae–bacteria groups: (**a**) algae group without the addition of C_6_-HSL; (**b**) algae group with the addition of C_6_-HSL; (**c**) algae–bacteria group without the addition of C_6_-HSL; (**d**) algae–bacteria group with the addition of C_6_-HSL.

**Figure 9 ijerph-20-05047-f009:**
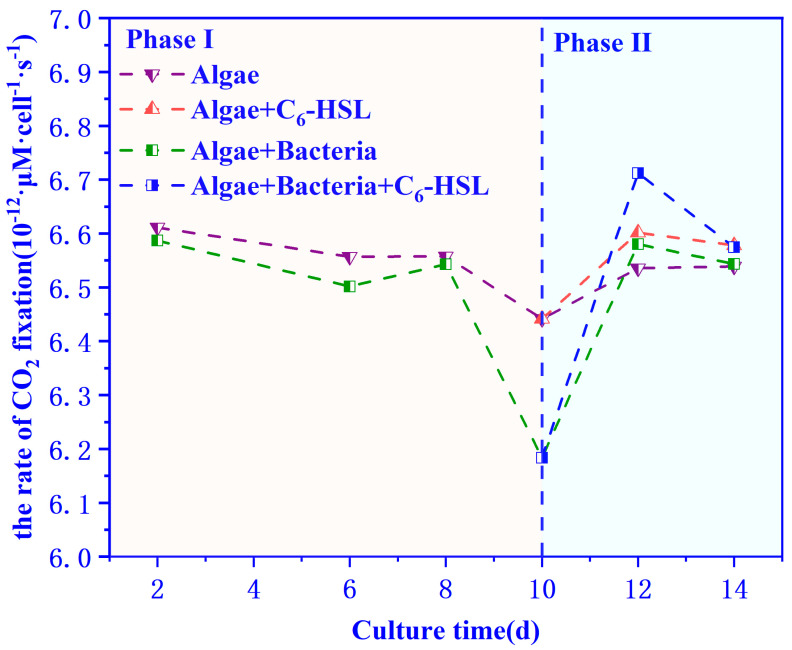
Effect of 10 ng/LC_6_-HSL on carbon fixation rates of algae group and algae–bacteria group. Algae+C_6_-HSL and algae+bacteria+C_6_-HSL are algae group and algae–bacteria group treated with C_6_-HSL, respectively.

**Figure 10 ijerph-20-05047-f010:**
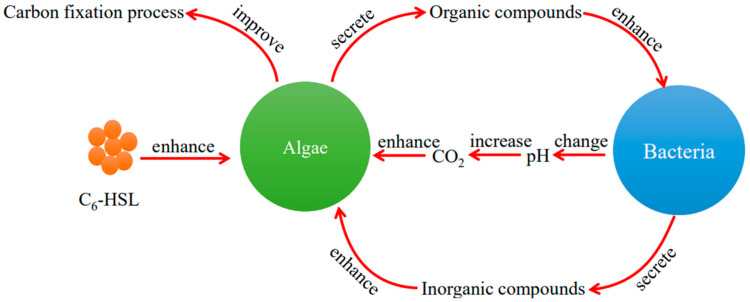
The mechanism of the effect of C_6_-HSL on the algae–bacteria system.

## Data Availability

Not applicable.

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
