# Peer review of "Effect of the N-hexanoyl-L-homoserine Lactone on the Carbon Fixation Capacity of the Algae–Bacteria System"

_ijerph, 2023, doi:10.3390/ijerph20065047_

Round 1
Reviewer 1 Report
The manuscript "Effect of the N-hexanoyl-L-homoserine lactone on the Carbon Fixation Capacity of the algal+bacteria system" investigated the effects of N-hexanoyl-L-homoserine lactone on algal+bacteria system and carbon fixtion efficiency. This topic is meaningful. However, the manuscript needs improvement to analyze the intrinsic cause for the experiment phenomenon. And some crucial scientific problems need more evidence or analysis. So, I suggest a minor revision for this manuscript in current form.
1. At the first paragraph and the second paragraph of the Introduction, there are too many common contents, and the research on the transformation of carbon source and sink state of carbon dioxide caused by algal + bacteria system is not summarized, which should be supplemented appropriately.
2. In the third to fourth paragraph of the introduction,the authors did not summarize the relationship between N-hexanoyl-L-homoserine lactone, algal + bacteria system, and CCM mechanisms, as well as existing studies, and major adjustments are needed.
3. At the experimental design part, what is the reason or basis for setting the AHLs concentration?
4. In paragraph 2.3.3, the equations (2)- (4) need to be supplemented by references.
5. In paragraph 2.3.4, the equations (5) also need to be supplemented by references.
6. Why is Staphylococcus ureilyticus strain CK27 symbiotic with Microcystis aeruginosa? Does it provide nutrients for Microcystis aeruginosa, or does it secrete some metabolites to promote the growth of Microcystis aeruginosa?
7. Microcystis aeruginosa (FACHB-905) can secrete toxins. Does AHLs promote carbon sequestration of algal + bacteria by affecting the content of toxins secreted by algae?
8. The language in the discussion part is somewhat verbose and needs to be concise.
9. Authors should provide high-resolution images of mechanism diagram in discussion part.
10. Algal is adjective, please use algae group and algae-bacteria group.
Reviewer 2 Report
This manuscript “Effect of the N-hexanoyl-L-homoserine lactone on the Carbon Fixation Capacity of the algal+bacteria system” is a useful and informative research in the field of related subjects. However, I still have some minor suggestions for the manuscript which are mentioned below:
1.The English of the manuscript should be improved.
2.The theme of this manuscript is the effect of the N-hexanoyl-L-homoserine lactone on the Carbon Fixation Capacity of the algal+bacteria system. But the introduction describes more about the types and mechanisms of carbon use by algae. Please shorten the introduction and focus on the algal+bacteria system.
3. In paragraph 2.1, the initial AHLs concentration used in Exp.1 need to be supplemented by references.
4. In paragraph 2.1, the medium used in this paper need to be supplemented by references.
5. The method of bacteria isolation isn’t mentioned. Please improve this and the method need to be supplemented by references.
6. All equations used in this paper need to be supplemented by references.
7. The parameters of Fridyanld model (CCM model) should be supplied.
8. In the Figure.6, the activity of CA enzyme and Rubisco enzyme should be a single figure and put in the paragraph 3.2.
9. Please elaborate on the future research needs in this domain.
